# A Study on the Levels of Selected Proangiogenic Proteins in Human Tissues and Plasma in Relation to Brain Glioma

**DOI:** 10.3390/ijms26104802

**Published:** 2025-05-16

**Authors:** Zuzanna Zielinska, Julia Giełażyn, Zofia Dzieciol-Anikiej, Janusz Dzieciol, Piotr Mrozek, Joanna Reszec-Gielazyn, Ewa Gorodkiewicz

**Affiliations:** 1Doctoral School, Department of Physical Chemistry, Faculty of Chemistry, University of Bialystok, Ciolkowskiego 1K, 15-245 Bialystok, Poland; z.zielinska@uwb.edu.pl; 2Higher School of Social Psychology, SWPS University, 03-815 Warsaw, Poland; 3Biobank, Medical University of Bialystok, Waszyngtona 13, 15-269 Bialystok, Polandjoannareszec@gmail.com (J.R.-G.); 4Department of Medical Pathology, Medical University of Bialystok, Waszyngtona 13, 15-269 Bialystok, Poland; 5Faculty of Mechanical Engineering, Bialystok University of Technology, Wiejska 45 C, 15-351 Bialystok, Poland; p.mrozek@pb.edu.pl; 6Department of Physical Chemistry, Faculty of Chemistry, University of Bialystok, Ciolkowskiego 1K, 15-245 Bialystok, Poland

**Keywords:** glioma, liquid biopsy, SPRi biosensor, biomedical research

## Abstract

Brain glioma is one of the most common malignant tumors of brain tissue. It is characterized by rich vascularization, which indicates the significant participation of angiogenesis in its growth and development. In its first stages, the disease is very often asymptomatic, and late diagnosis significantly limits possibilities of treatment. Tumor angiogenesis, i.e., the formation of new vessels, requires the presence of angiogenic compounds that will enable tumor progression by creating a path for the supply of nutrients. The proangiogenic compounds involved in the development of glioma include hypoxia-inducible factor 1α (HIF-1α), angiopoietin-2 (ANG-2), and interleukin-1β (IL-1β). The aim of this study was to analyze changes in the levels of these proteins in plasma samples of patients diagnosed with brain glioma in stages G1 to G4, and in a control group, using SPRi biosensors. The results obtained in plasma were compared with the concentrations obtained during the analysis of tissue homogenates from patients with glioma in stages G2 to G4. A statistically significant difference in plasma concentrations was obtained between the patient group and the control group. The concentrations of the markers in tissue homogenate samples were statistically higher than in blood plasma. There was no significant effect of gender, diet, smoking, or the patient’s general health condition (Karnofsky score) on the course of the disease. These factors do not directly increase the risk of developing brain glioma.

## 1. Introduction

Brain gliomas are currently the most common malignant tumors occurring in the human central nervous system (CNS). They account for approximately 70–80% of all tumors occurring in brain tissue [1]. According to the most straightforward classification, we can distinguish ependymomas, astrocytomas, oligodendrogliomas, and malignant gliomas [2]. The World Health Organization (WHO) classifies glioblastoma multiforme (GBM) as grade 4 glioma; this is the most common type of malignant brain tumor and is highly aggressive [3]. About 7% of patients survive for 5 years, and most patients die in the first 1.5 years of treatment. GBM can be subclassified as primary and secondary, based on the clinical picture. Primary tumors occur in elderly patients, while secondary tumors account for about 10% of cases, typically in young patients [4]. The most common type of glioma is astrocytoma. These are divided into multiple categories, including diffuse subtypes—pilocytic astrocytoma (WHO grade 1), subependymal giant cell astrocytoma (WHO grade 1), and pleomorphic xanthoastrocytoma (WHO grade 2 or 3)—as well as non-diffuse subtypes, which are classified based on mutations and histological features, according to the 2021 WHO classification system, as grade 2, 3, or 4. Grade 4 astrocytomas also include IDH-mutated astrocytomas and IDH-wild-type GBM. They can occur in both adults and children. Patients with IDH mutations exhibit a less aggressive course than those with IDH-wild-type gliomas. These mutations play essential roles in glioma and are also quite similar; they are therefore the target of many therapeutic approaches [2,5]. The next type, oligodendrogliomas, also grows in the frontal lobe; these are diffuse tumors and account for only 5% of CNS tumors. They are classified as grade 2 gliomas, which are minimally invasive, or as grade 3 in the case of the so-called anaplastic oligodendrogliomas, which are malignant and develop rapidly [6].

Because glioma grows infiltratively, and the tumor cells divide rapidly and migrate intensively, the tumor grows within the brain tissue, and complete surgical resection is difficult to perform. This leads to tumor recurrence. This tumor usually does not form metastases but shows a strong tendency toward local recurrence. Cases of distant spread of tumor cells through body fluids are rare [7].

The formation of new vessels strictly controls brain tumor progression and growth, and tumors located far from the vessels are not adequately oxygenated. This induces cancer stem cells to generate new blood vessels within the tumor—these vessels are larger and more irregular than normal vessels, tangled and highly branched [8,9]. Tumor angiogenesis occurs through the action of many angiogenic factors, which usually also participate in physiological angiogenesis, such as HIF-1α and angiopoietin-2 (ANG-2) [10]. Due to local swelling and inflammatory processes, pro-inflammatory factors, including interleukin-1β (IL-1β), also act in the area of the tumor.

Hypoxia-inducible factor (HIF-1) is a transcription factor produced in the body in response to hypoxia. It is a heterodimeric protein composed of alpha and beta subunits [11]. In physiological conditions, HIF-1α is degraded in the ubiquitin–proteasome pathway. In profound hypoxia, together with HIF-1β, it forms a complex that is a stable and functional transcription factor. The literature indicates that HIF-1α may contribute to the development of glioma by supporting the process of angiogenesis, increasing the formation of metastases, or stimulating the production of vascular endothelial growth factor (VEGF), which in turn contributes to tumor growth and the migration of tumor cells [12]. Angiopoietins also affect tumor angiogenesis. ANG-2 is an ANG-1 antagonist essential in forming new vessels and in tumor progression. Both are ligands of the tyrosine kinase receptor (Tie1 and Tie2), characteristic of endothelial cells. ANG-2 competitively binds with Tie2, which causes vessel destabilization and appears where remodeling of vessels is to occur. Therefore, it affects metastasis and the development of new tumors [13]. ANG-2 appears in cancer cells, while ANG-1 predominates in healthy tissues during natural physiological processes [14]. The last of the selected compounds, interleukin-1β, is a pro-inflammatory cytokine described in the literature as a modulator of glioma progression [15]. However, whether it helps or inhibits the development of glioma is not fully known. It is known that IL-1β activates endothelial cells to produce pro-angiogenic factors, but healthy tissues can also initiate apoptosis of cancer cells [16].

The Surface Plasmon Resonance method in the imaging version (SPRi) is an interesting tool due to its good sensitivity, label-free nature, and speed of biomarker detection [17]. Since the 1980s, when the first commercial sensor was developed, conventional SPR has been used in many fields, including diagnostics and the pharmaceutical and biomedical industries. SPR is also characterized by high sensitivity and performance over a wide dynamic range without significant electromagnetic interference [18].

Surface Plasmon Resonance uses plasmons excited at a metal surface. Surface plasmons have been observed in small metallic particles during electrical resonance excited by a photon [19]. The photon energy is absorbed by free electrons of a thin metal layer (e.g., gold or silver) and causes the excitation of electrons at the boundary between the dielectric and metal [20]. SPR is therefore a physical phenomenon at the boundary between two media, caused by a beam of light.

Photons of light fall on the metal layer, transferring energy to electrons, and the light is reflected within a circular prism, which results in total internal reflection. This enables the quantitative recording of biomaterial adsorption at the boundary, because surface plasmons excited at the metal–dielectric boundary cause a change in the intensity of the reflected beam. In this area, conditions of high sensitivity to changes occurring at the metal–dielectric boundary are created, such as the adsorption of molecules on the metal surface or the immobilization of molecular layers. This enables the detection of minimal amounts of material binding due to very high sensitivity to small changes in the refractive index near the surface [17,21]. Commercially available SPR devices use measurements of changes in the refractive index on a gold surface [22]. This can be modified with layers of thiols, which are efficient substrates for immobilizing particles. Conventional SPR differs from SPRi in that sensograms are plotted with SPR. The imaging version uses a CCD camera, where the results are recorded as an image and converted using mathematical operations into a measurable analytical signal [18,23]. Surface Plasmon Resonance can provide information on, among other things, the speed of interactions on the surface—namely, association and dissociation—making it possible to study kinetic rate constants and concentration changes. It also supplies data on binding and affinity constants that may be used in qualitative or semi-quantitative analyses. The determined kinetic constants offer information about the dynamics of the biological system being studied, and this can be used to select and rationally design new molecules of therapeutic importance. The combination of the SPR method with an appropriate arrangement of receptor layers, which leads to the construction of SPRi biosensors, opens up possibilities for identifying types of bonds and affinities in real time, for rapid analyses of association and dissociation constants, and for the study of the concentrations of immobilized layers [24]. On the surface of the gold chip, SAMs of various thiols can be produced, which enables immobilization of the sensor’s receptor layer. Possible thiols include cysteamine [25] and 11-mercaptoundecanoic acid (11-MUA), as used in this work. The layer immobilization procedures differ depending on the labeled particles and the receptor layers used [26].

In this work, matrix SPRi biosensors are used as a new analytical method to examine the levels of selected proteins (HIF-1α, ANG-2, and IL-1β) in plasma and tissue homogenates from patients with diagnosed brain glioma. Samples from patients with four stages of the disease and a control group were analyzed. The study aimed to find the relationship between the concentrations of proangiogenic factors and the stages of the disease. The correlations between the concentrations of the tested proteins and the parameters available in the clinical description of the samples were also studied. The results obtained were considered in the light of findings previously reported in the literature. In our laboratory, the VEGF-A and FGF-2 pathways were studied as factors involved in angiogenesis. Due to the association of the proteins selected in this article with the VEGF pathway, it was decided to examine them as distinct proangiogenic proteins in a separate study [27].

## 2. Results

The first stage of the study began with a test of the normality of the data distribution. The Shapiro–Wilk test, used in each case of changes in the concentrations of individual brain glioma biomarkers in plasma and tissue homogenates from patients, indicated non-normal distributions. For this reason, nonparametric tests were used for further analysis.

### 2.1. Angiogenic Compounds in the Blood Plasma of Glioma Patients

More than two data groups were compared using the Kruskal–Wallis ANOVA test. Figure 1 presents graphs of changes in protein concentrations—HIF-1α, angiopoietin-2, and interleukin-1β—according to the grade of brain glioma and the distribution of concentrations in blood plasma from the control group. In each of the three analyses, a statistically significant difference in concentrations was obtained between the group of patients and the control group (*p* < α). The post hoc Dunn–Bonferroni test indicated a lack of statistically significant differences in concentrations when comparing grades G1, G2, G3, and G4 of brain glioma. Statistically significant differences occurred for each of the three potential biomarkers when grades G2, G3, and G4 were compared with the control group. The graphical post hoc Dunn–Bonferroni analysis is presented in Figure 2. In general, HIF-1α levels were highest in the G2 glioma group and lowest in the control group. Angiopoietin-2 was present at the highest level in G3 glioma and at the lowest level in the control group. The lowest levels of IL-1β were also found in the control group, and in grades G1 and G2, its concentration remained constant. It was present at the highest level in the G4 glioma group. More detailed results of the analysis are presented in Table A1 in Appendix A.

Correlations between the studied proteins were determined using the Spearman monotonic relationship. This allowed potential specific relationships to be identified between levels of the HIF-1α, ANG-2, and IL-1β proteins in brain glioma patients. Control samples were omitted from the data so that they did not affect the relationships obtained. The analysis indicated a statistically significant, low positive correlation between HIF-1α and ANG-2 in patients with the disease. A positive correlation was found according to the Guilford scale, which means that an increase in the concentration of one of the potential proteins causes an increase in the concentration of the other. The results of the Spearman monotonic relationship analysis are presented in Table 1.

Spearman’s monotonic relationship analysis was also used to examine relationships between variables such as age, Karnofsky performance score, physical activity, amounts of vegetables, fruit, and meat in the diet, pack-years, the size of the tumor, and the concentrations of the tested proteins. The analysis included patients aged 29 to 77. The Karnofsky Performance Scale serves to evaluate the general condition and quality of life of a patient living with cancer and referred for chemotherapy or radiotherapy. The scale includes specific criteria, and a score of 100 denotes an ideal condition, while 0 denotes the patient’s death. Among the study participants, there were four persons with a score of 100 (without any symptoms or illnesses), 16 with a score of 90 (regular activity and minor symptoms and illnesses), 20 with a score of 80 (a state of almost complete activity, with minor symptoms and illnesses), and 11 with a score of 70 (a state without the ability to work and a moderate condition). One person required periodic care but had the ability to perform most of their daily needs independently (60), and one person required frequent care and frequent medical interventions (50). Physical activity was described on a 3-point scale, with 3 denoting intense physical activity (9 patients), 2 denoting moderate physical activity (42 patients), and 1 denoting no physical activity (2 patients). Dietary vegetable and fruit consumption was analyzed on a scale from 1 to 7, with 7 denoting daily consumption (35 patients), 6 denoting consumption several times a week (13 patients), 5 denoting consumption once a week (4 patients), and 4 denoting consumption once a month (1 patient). There were no patients at levels 1–3. The amount of meat consumed was expressed as a number of days per week: 37 patients ate meat 7 days a week, 6 patients did so 6 days a week, 4 did so 5 days a week, 5 did so 3 days a week, and 1 patient ate meat 1 day a week. Pack-years are used in medicine as an indicator of the risk of developing diseases related to tobacco smoke. They are calculated by multiplying the number of packs of cigarettes smoked per day by the number of years of addiction. The results of the relationship analysis are presented in Table 2.

A statistically significant moderate negative correlation exists between HIF-1α concentration and patient age. This indicates that higher HIF-1α concentrations occur in younger patients. A statistically significant low negative correlation exists between IL-1β concentration and physical activity (higher IL-1β concentrations were found in patients with low physical activity). The Mann–Whitney U test was used to examine other variables. It was investigated whether gender, alcohol consumption, coexisting non-cancer diseases, and cancer in the family history affect the concentrations of the tested proteins. A statistically significant result was obtained only for HIF-1α levels depending on whether there was cancer in the family history. The concentrations of this protein are higher in people with a family history of cancer. The results of the analysis are presented in Figure 2.

The diagnostic usefulness of the constructed sensors was tested using ROC curves. Strong statistical significance was obtained; the curves are presented in Figure 3. Data from the ROC curves are given in Table 3. For each tested protein (HIF-1α, ANG-2, and IL-1β), an increase in concentration implies an increase in the chances of developing brain glioma. This indicates that concentration values equal to or higher than a given cut-off point (expressed in concentration units) may be taken to indicate the presence of the disease. The area under the ROC curves (AUC) determines the accuracy of the assignment of patients to the sick or healthy group. The closer the AUC value is to one, the more precisely it is known that the assignment of patients to particular groups is correct. In the case that AUC = 0.5, it can be stated that there is random assignment to the groups. In the analysis, the AUC value for all three markers was very close to one (Table 3), meaning that the distribution of patients between the groups was not random. Table 4 gives values of sensitivity and specificity, which are also good diagnostic test assessments. The test demonstrated the best sensitivity for IL-1β (94.34%), followed by ANG-2 (88.68%) and HIF-1α (84.91%). Biosensors also exhibited adequate specificity: the probability of detecting a potentially healthy person was 95.83% in the case of HIF-1α, 85.42% in the case of ANG-2, and 77.08% in the case of IL-1β. Positive predictive value (PPV) indicates the probability that a person with a positive test result has a brain glioma, while negative predictive value (NPV) indicates the probability of the absence of the disease in the case of a negative test result. Thus, for example, when testing is based on HIF-1α, a person with a positive test result has a 95.74% probability of having brain glioma. On the other hand, the probability of not having the disease in case of a negative test result is 85.19%.

### 2.2. Determination of HIF-1α, ANG-2, and IL-1β in Tissue Homogenates from Patients with Brain Glioma

As in the case of plasma samples, the Kruskal–Wallis ANOVA test was performed. This confirmed a statistically significant difference in concentrations between the grades of brain glioma only in the case of IL-1β. Post hoc Dunn–Bonferroni analysis indicated a significant difference in IL-1β levels between grades G2 and G3. The graphical analysis is presented in Figure 4, with graphs for HIF-1α, ANG-2, and IL-1β. More detailed results of the analysis are presented in Table A2 in Appendix A.

Spearman’s monotonic relationship analysis revealed a lack of correlation between the proteins studied. However, an examination of the previously described variables revealed some relationships. The results are presented in Table 4. There is a statistically significant low negative correlation between IL-1β concentration and patient age (lower IL-1β concentrations were obtained in samples from older patients). In addition, a statistically significant low negative correlation was obtained between HIF-1α concentration and physical activity (higher concentrations of this protein occur in samples from people with low physical activity). A high positive correlation was obtained between HIF-1α level and tumor size. The higher the HIF-1α levels in the tissue sample, the greater the size of the tumor. This confirms the theory that in hypoxic conditions, i.e., during tumor growth and development, HIF-1α is released, supporting further growth of the pathological change.

The Mann–Whitney U test showed no statistically significant differences in the concentrations of the angiogenic proteins studied depending on the patient’s gender. The results of the analyses are presented in Figure 5. A statistically significant difference was obtained in concentrations of IL-1β between the group of patients with coexisting non-neoplastic diseases and those without such diseases. IL-1β concentrations were higher in samples from patients with diagnosed coexisting non-neoplastic diseases.

### 2.3. Comparison of HIF-1α, ANG-2, and IL-1β Levels in Plasma Samples and Tissue Homogenates

The control group and grade G1 were excluded from the comparison due to the lack of tissue homogenate samples for those groups. The Shapiro–Wilk test showed that the data distribution differed from normal, and therefore, the analysis was conducted using nonparametric tests. The Kruskal–Wallis ANOVA test showed statistically significant differences between the groups for each tested protein. The distribution of concentrations is presented in Figure 6.

Post hoc Dunn–Bonferroni analysis was used to determine which specific groups had significant differences. The graphical analysis is presented in Figure 7. For the analysis, it was essential to compare the same glioma grade in different biological materials. For HIF-1α and IL-1β, significant differences occurred between plasma and tissue homogenate samples in grades G2 and G4. For ANG-2, statistically significant differences were observed only in grade G4. Comparing the concentrations, for HIF-1α, a higher concentration was obtained in plasma samples in each grade. In the case of ANG-2 and IL-1β, the average concentrations were higher in tissue homogenate samples.

## 3. Discussion

This paper presents changes in the concentrations of the tested proangiogenic proteins HIF-1α, ANG-2, and IL-1β in blood plasma samples and tissue homogenates in various degrees of malignancy of brain glioma. The influences of the patient’s habits, family history, or the presence of other diseases on the development of the brain tumor were also examined. The results obtained using plasma concentrations were compared with those for the control group (CTR). Intercorrelations between the tested proteins were also investigated.

Statistically significant differences were obtained in the plasma concentrations of all three proteins between patients and the control group (Figure 1). This is indicated by the *p*-value when the control group was compared with the group of patients with various degrees of glioma as a whole. The concentrations in the control group were significantly lower than in each degree of brain glioma, excluding grade G1. The Dunn–Bonferroni post hoc test indicated statistically significant differences only between the G2, G3, and G4 groups and the control group. At the G1 stage, there is no significant difference compared with the control. Increased levels of proangiogenic proteins during the disease can be explained by their participation in the angiogenesis process, which is responsible for glioma invasion and disease development [28]. We also observed that the highest concentrations of HIF-1α occur in stage G2. Studies indicate that the expression of hypoxia-inducible factor is associated with tumor development and progression, and so in stage G2, a high level of this protein may be expected [29]. Its level decreases in stages G3 and G4, where patients are already receiving appropriate treatment. Lower concentrations of HIF-1α were obtained in tissue homogenate samples; these differences were statistically significant in the case of G2 and G4. The highest level of ANG-2 in blood plasma was observed in grade G3.

During the progression of a low-grade tumor to a higher-grade glioma, remodeling occurs at the level of the vascular phenotype. Angiopoietin-2 correlates with an increase in malignancy and changes in the vascular phenotype [30]. The highest ANG-2 level was observed in grade G4 in the tissue homogenate sample. ANG-2 exhibits high expression even in the first stages of the disease; these concentrations are much higher than in the control. It can therefore be identified as an early marker of brain glioma.

Interleukin-1β, as a proinflammatory cytokine, is expected to be present in the tumor environment, and according to the literature, is also produced by the tumor. We observed lower concentrations in the control group and in grades G1–G3, while high concentrations were found in grade G4 (highly malignant tumor). Studies indicate that cells do not produce large amounts of IL-1β at low grades, and expression increases with increasing tumor malignancy [31]. The results of this study are therefore in agreement with the literature. A different situation was observed in the case of determinations in tissue homogenates. Apart from the fact that the concentrations are higher in each grade, which was predictable due to the samples’ being taken from the tumor mass, the medians indicate that the concentrations are the highest in grade G2. Due to the small number of grade G3 samples, we cannot unequivocally confirm the obtained results. Nevertheless, the results on many plasma samples indicate the diagnostic value of IL-1β as a marker of brain glioma.

An examination of the correlation between proteins also provides some valuable information. Plasma studies indicate a low positive correlation between HIF-1α and ANG-2, suggesting that an increase in HIF-1α concentration causes a subsequent rise in ANG-2 activity. This can be explained by the fact that hypoxia occurs in the tumor environment during tumor growth, so that HIF-1α activity increases. To supply the tumor with the appropriate nutrients in response to the lack of oxygen availability, cancer cells induce the formation of a new blood supply to the tumor, i.e., a new blood vessel. We know that angiopoietin-2 appears where vascular remodeling occurs, and a higher intensity of this process is associated with its increased expression [13]. The increase in ANG-2 concentration in response to the appearance of HIF-1α is also justified. No correlation between proteins was observed in the case of tissue homogenate samples.

To test the influence of the patient’s living environment and habits, such parameters as age, Karnofsky Performance Scale, physical activity, the presence of meat, vegetables, and fruits in the diet, the frequency of smoking, and the size of the tumor were taken into account. Some studies indicate that people with a sedentary lifestyle, having an improper diet, drinking alcohol, and smoking cigarettes have a higher risk of developing brain tumors [32]. We therefore decided to test whether the obtained levels of proangiogenic compounds in the case of brain glioma confirm the association of that disease with the aforementioned factors. An analysis of the results for plasma samples showed that the variables had no significant effect; however, a negative moderate correlation was obtained for HIF-1α and age. Higher concentrations of this protein are found to occur in younger patients, but this has no relevance and has no significant effect on the risk of developing brain glioma. In addition, the low negative correlation of IL-1β with physical activity indicates that higher concentrations of IL-1β occur in patients with lower physical activity. The literature supports this thesis: physical activity promotes the release of interleukin-6 from muscles, which acts as a myokine and induces an anti-inflammatory response through the secretion of interleukin-10 and the inhibition of IL-1β [33]. However, it has no direct effect on brain glioma. Despite the existence of weak correlations, the results indicate that within this study group, there was no significant effect of age, diet, or physical activity on the concentrations of the determined proteins, and these factors did not cause an increase in the incidence of glioma. An interesting result may be the strong positive correlation between HIF-1α and tumor size for samples taken from the tumor mass (homogenates). The larger the tumor, the higher the concentration of hypoxia-inducible factor. We already know that HIF-1α regulates the action of vascular endothelial growth factor (VEGF), which is probably the primary mediator of angiogenesis during malignant gliomas. HIF-1α expression affects glioma tumor growth, which suggests clinical applications in treating malignant gliomas [34].

The results for both biological fluids indicate that gender does not affect the levels of the studied proteins. However, HIF-1α occurs at statistically significantly higher levels in the plasma of patients with a family history of cancer. In the case of some other cancer diseases, such as breast cancer, it has been shown that HIF-1α plays a dominant role in the development of cancer angiogenesis. This applies to familial cancers, i.e., BRCA1-2 gene carriers. Elevated levels of HIF-1α may therefore impact the development of other cancers [35]. IL-1β was found in higher concentrations in homogenate samples when in the presence of other non-cancerous diseases. This finding seems logical in view of the presence of IL-1β in the body in the case of inflammation. A comparison of concentrations within the same glioma grades in plasma samples and homogenates showed statistically significant differences in the case of HIF-1α in grades G2 and G4, for ANG-2 in grade G1, and for IL-1β in grades G2 and G4. The relationship between groups was not analyzed. As previously established, these differences are predictable—the homogenate sample is taken from the tumor mass, where the concentrations of the tested proteins may be expected to be higher.

Due to the lack of a control group for homogenates, ROC analysis was performed only for the results from plasma samples. Good sensitivities, specificities, PPV, and NPV were obtained for each tested protein. Tests for glioma based on the determining concentrations of HIF-1α, ANG-2, and IL-1β will indicate the presence of the disease at higher concentrations of these proteins, because the diagnostic value of each variable increases with concentration.

## 4. Materials and Methods

### 4.1. Chemical Reagents and Materials

The following reagents were used for the construction of the biosensor and validation of the analytical method: recombinant HIF-1α protein, recombinant human ANG-2 protein, and recombinant human IL-1β protein; monoclonal mouse antibody against HIF-1α, monoclonal mouse antibody against ANG-2, and monoclonal mouse antibody against IL-1β (all from R&D Systems, Minneapolis, MN, USA); EDC (N-ethyl-N′-(3-dimethylaminopropyl)carbodiimide hydrochloride) (SIGMA, Steinheim, Germany); 11-MUA 11-mercaptoundecanoic acid (Aldrich, Munich, Germany); NHS N-hydroxysuccinimide (Aldrich, Munich, Germany); buffered saline solution (PBS buffer) (Biomed, Lublin, Poland); absolute ethyl alcohol (POCh, Gliwice, Poland); and ethanolamine solution (SIGMA, Steinheim, Germany). The base of the biosensor is a plate with a gold layer (Ssens, Enschede, The Netherlands).

### 4.2. SPRi Device and Analysis

The research was carried out using the Surface Plasmon Resonance imaging (SPRi) equipment available at the Bioanalysis Laboratory, Faculty of Chemistry, University of Białystok. The structure of the analyzer and the layers produced on the surface of the biosensor are shown in Figure 8. The measurement system consists of polarizers and lenses through which the light emitted by an LED laser passes before falling on the prism with the biosensor. There, the light beam (wavelength λ = 635 nm) is reflected, and the light is collected by a CCD camera, where it is processed into an image. A quantitative signal is then obtained through mathematical operations.

The biosensor plate is made of BK7 glass, on which a titanium layer (1 nm) is sputtered to facilitate the adsorption of the next layer—gold (50 nm). The chip is immersed in a 1 mM alcoholic solution of thiol (11-mercaptoundecanoic acid) for 24 h. After washing and drying in an argon stream, a polymer foil is applied, which isolates nine measurement sites. A 1:1 mixture by volume of EDC (0.4 M) and NHS (0.1 M) is applied to the active sites for 10 min. After removing the excess solution, a specific antibody is applied. After another 10 min, ethanolamine is applied to the active sites to prevent non-specific adsorption of other particles. The chip is ready for measurement after washing with PBS buffer and collection of excess solution. It is applied to the prism using immersion oil, and the appropriate angle is selected using the analyzer’s movable arms. The first measurement is then performed. The next step is to apply the samples to the sites where the antibody has been immobilized. A sample volume of 3 microliters is used for testing. After 10 min, another measurement is performed after washing the excess solutions with PBS buffer. The analytical signal, therefore, is the difference in the intensity of the light reflected before and after interaction with the analyte.

Calibration curves plotted during the construction of biosensors sensitive to HIF-1α, ANG-2, and IL-1β were used for the studies [36]. When plotting the saturation curves, the optimal antibody concentrations were selected: 50 ng/mL for the HIF-1α-specific antibody, 5 μg/mL for the ANG-2-specific antibody, and 100 ng/mL for IL-1β. The plasma and homogenate samples were diluted appropriately to fall within the linear concentration range of the biosensor–dilutions are shown in Table 5. The linear ranges and calibration curves are shown in Section A.3.

### 4.3. Biological Material

The study used 54 plasma samples from patients diagnosed with brain glioma (3 samples of grade G1, 10 of grade G2, 6 of grade G3, and 35 of grade G4). The control group consisted of plasma from 48 patients who smoked cigarettes but had not been diagnosed with cancer. Samples were taken before surgery. The samples came from the Biobank of the Medical University of Białystok, and consent to conduct the study was obtained from the relevant bioethics committee (approval R-I-002/600/2019, 19 December 2019). The study also included measurements on 32 tissue homogenate samples (9 samples of grade G2, 2 samples of grade G3, 21 samples of grade G4). Biological material (brain tumor tissue) was collected from patients diagnosed with brain glioma (grades G1–G4). It was collected in 2017–2018 and stored in the Biobank of the Medical University of Białystok (MUoB) at −176 °C in liquid nitrogen vapor. The relevant Bioethics Committee of MUoB approved the conduct of research using this material (approval APK.002.171.2021, 25 March 2021). The homogenate samples were taken from the same patients as the plasma samples. However, we did not obtain equal groups: not every plasma sample had a corresponding homogenate available, and similarly, not every homogenate sample had a plasma sample from the same patient.

#### Homogenates and Plasma Sample Preparation

Tumor tissue samples were standardized—200 mg of tissue cut into small pieces was placed in a tube, and then, 2 mL of ice-cold PBS buffer mixed at a ratio of 1:100 with a cocktail of protease inhibitors (10 μL of inhibitor solution per 1 mL of PBS) was added. The filled tubes were then placed on ice and homogenized using ultrasound at 20 kHz for 10 cycles (each cycle lasted for 60 s; intervals between cycles were 2 s). Next, the samples were centrifuged at 15.093× *g* for 10 min. Then, 1 mL of supernatant was collected in another tube and stored at −80 °C. Venous blood was collected in a closed system with ethylenediaminetetraacetic acid (EDTA) in vacuum tubes. Blood samples were centrifuged at 3000 rpm for 15 min (MPW-53 centrifuge). Then, the supernatant (plasma) was collected and transferred to polypropylene tubes (1 mL). Samples were stored at −80 °C.

## 5. Conclusions

The studies indicate statistically significant differences in the plasma concentrations of the tested proteins HIF-1α, ANG-2, and IL-1β between the patients and the control group. These differences are evident between the more advanced stages of the disease and the control group. The concentrations of proangiogenic proteins are higher for HIF-1α and IL-1β in tissue homogenates; ANG-2 remains at a similar level. Most of the variables tested do not correlate with the levels of these proteins, and if such correlations do occur, they are weak. A strong correlation was found only between the level of HIF-1α and the size of the tumor. This also represents a starting point for the clinical use of compounds inhibiting the activity of HIF-1α to help treat malignant gliomas. Undoubtedly, these three proteins are associated with the development of brain glioma and may be potential biomarkers of this disease. However, this method is unsuitable for making a precise diagnosis in early glioma detection, because higher levels are observed in more advanced stages.

It is unclear whether these proangiogenic factors determined in plasma are a product of glioma development. Other angiogenic factors, such as the previously studied VEGF-A [27], showed an increased level in the blood plasma of patients with brain glioma compared with the control group. We expected similar results when examining HIF-1α, ANG-2, and IL-1β in plasma. Nevertheless, they did not provide a precise diagnosis of glioma, and we wished to examine how these results for plasma relate to the protein levels obtained in homogenate samples. We expected the latter levels to be higher, but the analysis showed that the homogenate samples contained lower amounts of the tested proteins than plasma. However, plasma concentrations are elevated compared with the control group and indicate the presence of increased angiogenesis, which may also be due to brain glioma.

## Figures and Tables

**Figure 1 ijms-26-04802-f001:**
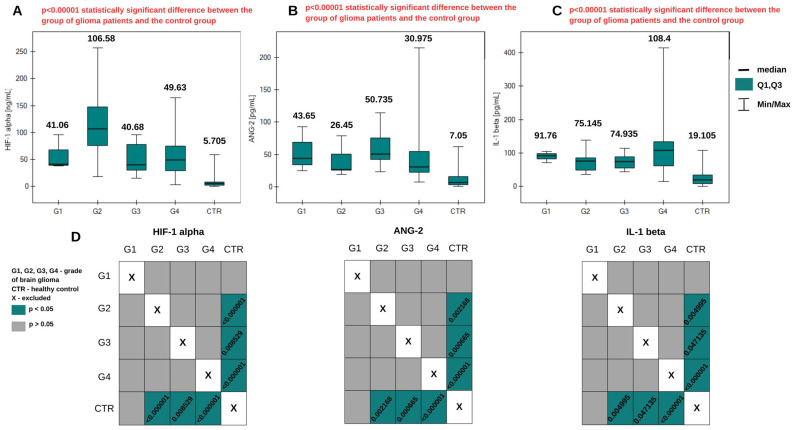
Graphs of changes in HIF-1α (**A**), ANG-2 (**B**), and IL-1β (**C**) concentrations in plasma samples from patients with brain gliomas of different grades. Information about the medians is placed above the box plots. (**D**) Dunn–Bonferroni POST-HOC graphical analysis. The green boxes with statistically significant values contain the *p* parameter values.

**Figure 2 ijms-26-04802-f002:**
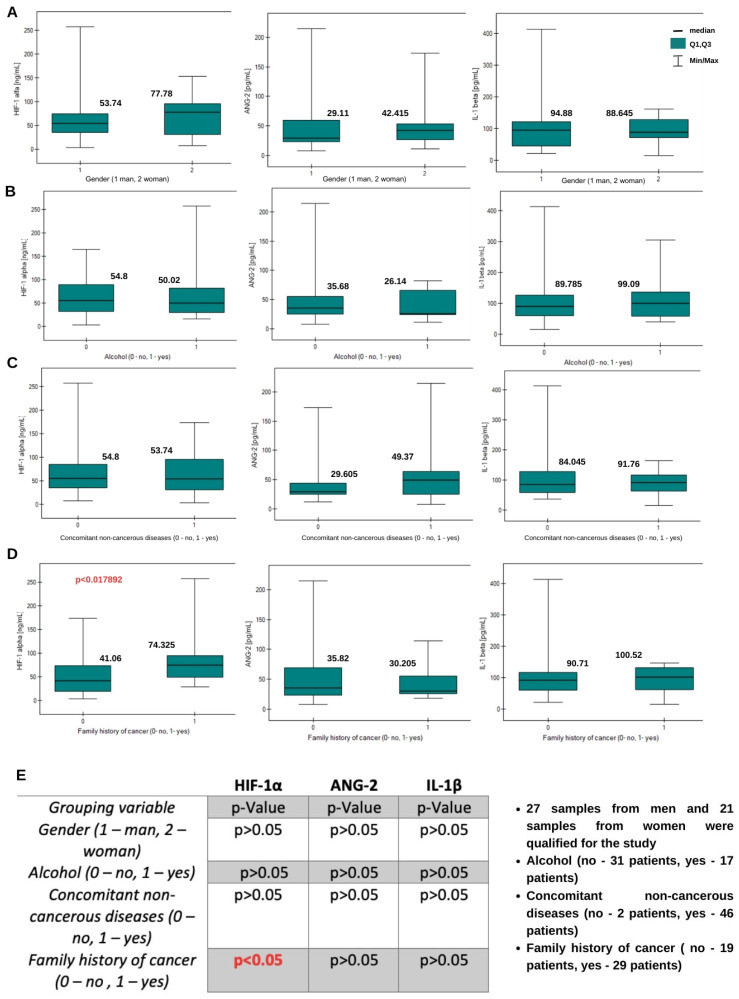
Graphs of the changes of HIF-1α, ANG-2, and IL-1β concentrations in plasma samples depending on (**A**) gender, (**B**) alcohol use, (**C**) concomitant non-cancerous diseases, and (**D**) family history of cancer. Information about the medians is placed above the box plots. (**E**) Mann–Whitney-U test *p*-value results table.

**Figure 3 ijms-26-04802-f003:**
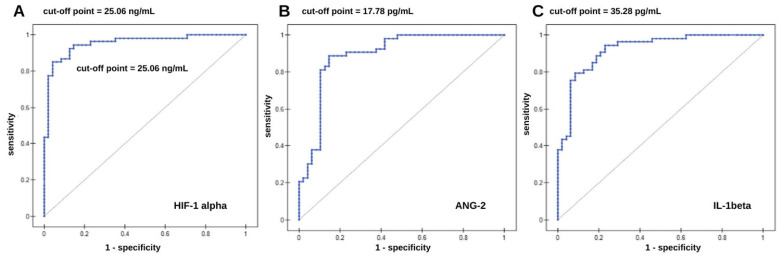
ROC curves for (**A**) HIF-1α; (**B**) ANG-2; and (**C**) IL-1β.

**Figure 4 ijms-26-04802-f004:**
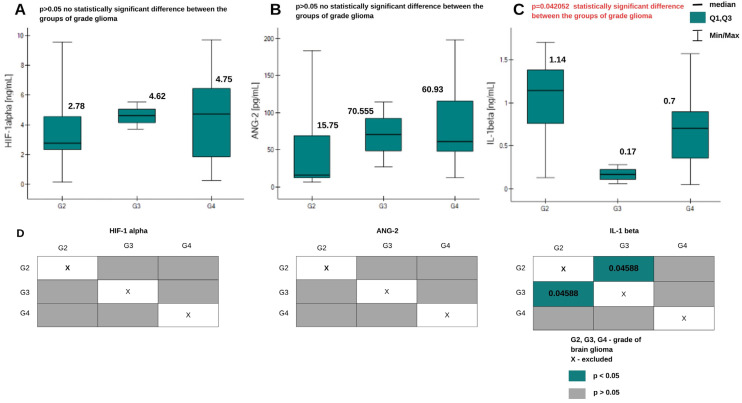
Graphs of changes in HIF-1α (**A**), ANG-2 (**B**), and IL-1β (**C**) concentrations in tissue homogenate samples from patients with brain glioma of different grades. Information about the medians is placed above the box plots. (**D**) Dunn–Bonferroni POST-HOC graphical analysis. The green boxes with statistically significant values contain the *p* parameter values.

**Figure 5 ijms-26-04802-f005:**
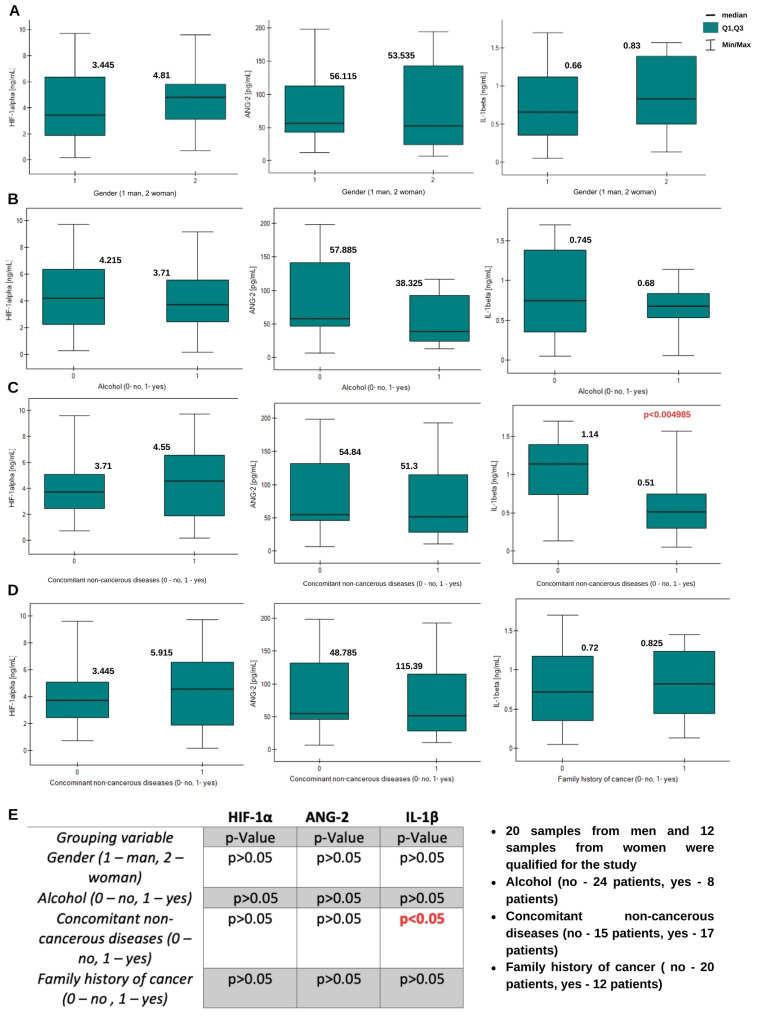
Graphs of changes in HIF-1α, ANG-2, and IL-1β concentrations in tissue homogenate samples depending on (**A**) gender, (**B**) alcohol use, (**C**) concomitant non-cancerous diseases, and (**D**) family history of cancer. Information about the medians is placed above the box plots. (**E**) Mann–Whitney U test *p*-value results table.

**Figure 6 ijms-26-04802-f006:**
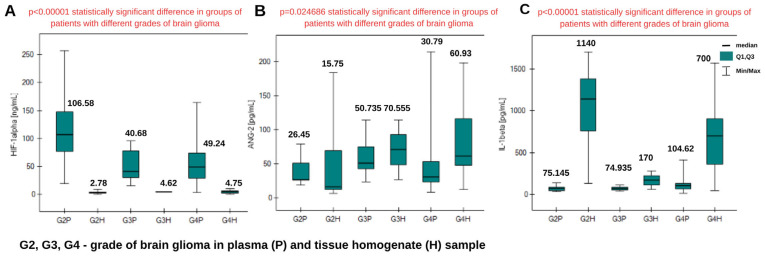
Graphs of changes in HIF-1α (**A**), ANG-2 (**B**), and IL-1β (**C**) concentrations in tissue homogenates and plasma samples from patients with brain glioma of different grades. Information about the medians is placed above the box plots.

**Figure 7 ijms-26-04802-f007:**
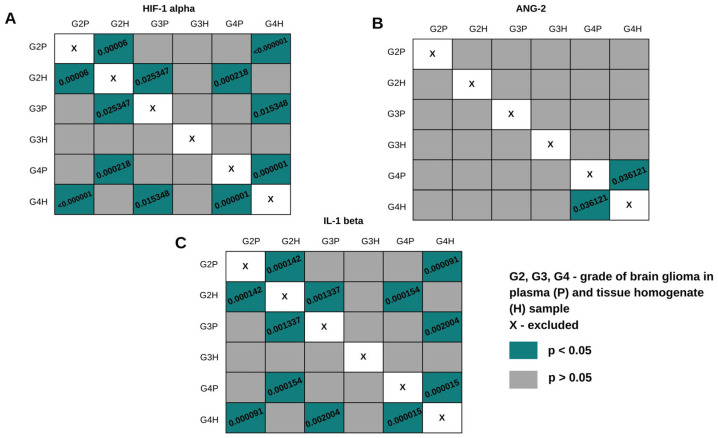
Dunn–Bonferroni POST-HOC graphical analysis in plasma and tissue homogenate samples in different grades of brain glioma. The green boxes with statistically significant values contain the *p* parameter values.

**Figure 8 ijms-26-04802-f008:**
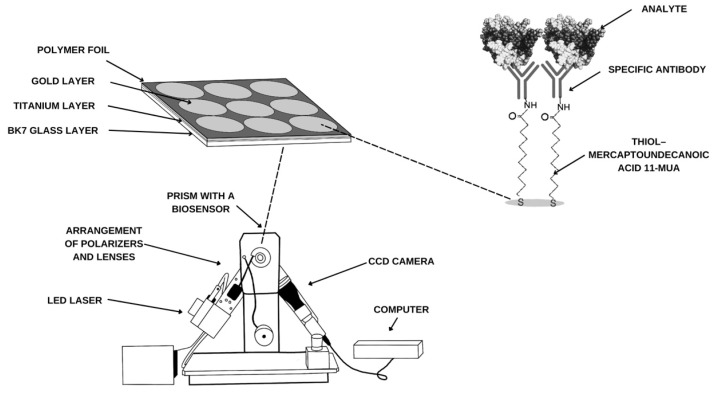
A schematic diagram of the SPRi apparatus and immobilization of layers on the biosensor surface.

**Table 1 ijms-26-04802-t001:** Correlations between determined proteins in human plasma samples with brain glioma. Interpretation of r coefficients based on J. Guilford’s scale.

Pairs of Proteins	Correlation Strength [r]	*p*-Value
HIF-1α–ANG-2	0.31 low positive correlation	*p* < 0.05
HIF-1α–IL-1β	0.02 no correlation	*p* > 0.05
ANG-2–IL-1β	−0.06 no correlation	*p* > 0.05

**Table 2 ijms-26-04802-t002:** Correlations between assayed proteins and other analyzed variables in plasma samples from patients with brain glioma. The interpretation of r coefficients based on the J. Guilford scale.

	HIF-1α		ANG-2		IL-1β	
Analyzed Variable	Correlation Strength [r]	*p*-Value	Correlation Strength [r]	*p*-Value	Correlation Strength [r]	*p*-Value
Age	**−0.41** **moderate negative correlation**	***p* < 0.05**	−0.007 no correlation	*p* > 0.05	0.23 no correlation	*p* > 0.05
Karnofsky Performance Scale	0.23 no correlation	*p* > 0.05	0.016 no correlation	*p* > 0.05	−0.201 no correlation	*p* > 0.05
Physical activity in the patient’s life	0.15 no correlation	*p* > 0.05	−0.003 no correlation	*p* > 0.05	**−0.28** **low negative correlation**	***p* < 0.05**
Vegetables and fruits in the diet	0.06 no correlation	*p* > 0.05	0.02 no correlation	*p* > 0.05	0.06 no correlation	*p* > 0.05
Meat in the diet	0.02 no correlation	*p* > 0.05	0.09 no correlation	*p* > 0.05	0.09 no correlation	*p* > 0.05
Pack-years	−0.13 no correlation	*p* > 0.05	−0.11 no correlation	*p* > 0.05	0.14 no correlation	*p* > 0.05
Size of the tumor	0.03 no correlation	*p* > 0.05	0.05 no correlation	*p* > 0.05	−0.12 no correlation	*p* > 0.05

**Table 3 ijms-26-04802-t003:** The data characterizing the ROC analysis. AUC—Area Under Curve; PPV—Positive Predictive Value; NPV—Negative Predictive Value.

Marked Protein	Direction of the Diagnostic Variable	AUC	Sensitivity [%]	Specificity [%]	PPV [%]	NPV [%]	Cut-Off Point	*p*-Value
HIF-1α	stimulant	0.95	84.91	95.83	95.74	85.19	25.06 ng/mL	*p* < 0.01
ANG-2	stimulant	0.89	88.68	85.42	87.04	87.23	17.78 pg/mL	*p* < 0.01
IL-1β	stimulant	0.92	94.34	77.08	81.97	92.50	35.28 pg/mL	*p* < 0.01

**Table 4 ijms-26-04802-t004:** Correlations between the assayed proteins and other analyzed variables in the tissue homogenate samples from patients with brain glioma. Interpretation of r coefficients based on the J. Guilford scale.

	HIF-1α		ANG-2		IL-1β	
Analyzed Variable	Correlation Strength [r]	*p*-Value	Correlation Strength [r]	*p*-Value	Correlation Strength [r]	*p*-Value
Age	0.03 no correlation	*p* > 0.05	−0.09 no correlation	*p* > 0.05	**−0.37** **low negative correlation**	***p* < 0.05**
Karnofsky Performance Scale	0.27 no correlation	*p* > 0.05	−0.24 no correlation	*p* > 0.05	0.29 no correlation	*p* > 0.05
Physical activity in the patient’s life	**−0.39** **low negative correlation**	***p* < 0.05**	−0.002 no correlation	*p* > 0.05	0.13 no correlation	*p* > 0.05
Vegetables and fruits in the diet	−0.10 no correlation	*p* > 0.05	−0.10 no correlation	*p* > 0.05	0.19 no correlation	*p* > 0.05
Meat in the diet	−0.02 no correlation	*p* > 0.05	−0.11 no correlation	*p* > 0.05	−0.30 no correlation	*p* > 0.05
Pack-years	−0.19 no correlation	*p* > 0.05	−0.09 no correlation	*p* > 0.05	−0.14 no correlation	*p* > 0.05
Size of the tumor	**0.72** **high positive correlation**	***p* < 0.05**	0.05 no correlation	*p* > 0.05	−0.12 no correlation	*p* > 0.05

**Table 5 ijms-26-04802-t005:** Dilution of samples for analysis.

Biomarker	Type of Biological Material	Dilution
HIF-1α	Plasma	10-fold
	Tissue homogenate	100-fold
ANG-2	Plasma	2-fold
	Tissue homogenate	10-fold
IL-1β	Plasma	10-fold
	Tissue homogenate	100-fold

## Data Availability

Data are contained within the article.

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
