# Peer review of "A Study on the Levels of Selected Proangiogenic Proteins in Human Tissues and Plasma in Relation to Brain Glioma"

_ijms, 2025, doi:10.3390/ijms26104802_

Round 1

Reviewer 1 Report

Comments and Suggestions for Authors

In the paper titled ‘Study of the level of selected proangiogenic proteins in human 2 tissues and plasma in relation to brain glioma’ Zielinska et al used SPR to detect levels of expression of three angiogenic factors in glioma samples.

I have e few concerns with this manuscript:

Angiogenesis in glioma, especially high-grade gliomas like glioblastoma, is driven by several genes and signaling pathways. The main ones are VEGF, FGF, angiopoietins, PDGF, NOTCH, What is the reasoning behind only studying HIF-1α, ANG-2 and IL- 25 1β

How much sample (blood and tissue homogenate) is loaded for the SPRi? At least for the homogenate is there any protein estimation performed prior to performing SPRi?

What is the actually values obtain for the significance analysis on Figure 2. The graphs appear quite noisy with heavy variation as discerned by the error bars. When were the blood samples collected
? Before or after surgery?

Not sure how the authors correlate diet and other variables to the levels of the angiogenic factors. Is there any literature supporting a connection? Authors should explain why they looked at these variables with regards to published works. Given than no strong correlation with any factors was found in table 3, one can only assume that its hard to make any associations?

What is the sample sizes used to generate figure 3? Use terminology Gender-male and female instead of sex man and woman. In general it is better to plot all the points along with the boxplots for such figures. Authors should do this for all figures.

In regards to the data of IL2, HIF1a and ANG2 presented in figure 5 (tissue homogenate) and figure2 (plasma), there is no concordance of results between the plasma and tissue samples? Which is seen in Figure7. Which brings me to the question of how the authors can use plasma as a surrogate for IL2, ANG2 and HIF1a secreted only from the glioma.. there can be angiogenesis process taking place in other parts of the human body..

Comments on the Quality of English Language

Extensive editing of English language required. Often the text is too colloquial.

Reviewer 2 Report

Comments and Suggestions for Authors

The ms by Zielinska et al. presents the analysis of the amount of selected proangiogenic proteins in gliomas. The research is reasonably planned and performed, using Surface Plasmon Resonance - not a new technique in general, but unique in neurooncology. The set of chosen proteins consists of only 3 items, but the authors sufficiently explained their choice. The material consisted of 54 plasma samples from glioma patients, 48 plasma samples from the control group, and 32 brain glioma tissue samples - the numbers are sufficient for the analysis. Some of the results are presented, and the "Discussion" (probably, named as 6. Results) combines them with other findings and existing knowledge.

My remarks to the ms are as follows:

1. The introductory part about gliomas has to be reviewed. It reveals that the authors are rather new in the neurooncological field. Some of the necessary changes are listed below:
line 41-42: there is no such information in the cited reference; most common neoplastic tumors in the brain are metastases
line 43: grades 1-4
line 43: IDH status is not the marker for glioma grading
line 46-47: "Despite [...] slightly" Although I understand the intention, that sentence is too much summary from lab to the clinic.
line 47-56: Needs revision, because the information included in the text relies on glioblastoma, or high-grade gliomas, not all gliomas as a group (which is very homogenous, as pilocytic astrocytoma G1 and GBM G4 cannot be compared)
line 104: [20,16]
line 246: Karnofsky Fitness Scale???

2. For comparison, the background for the method is described with professional knowledge. However, it is in general too narrative and can be made more formal and simple (e.g. line 92-94 "energy [...] is transformed into electromagnetic excitations in the form of charge density oscillations; line 95 "method can be called [...] phenomenon"?).

3. Figure 1. "own elaboration" - not necessary information

4. I cannot understand the choice of smokers for the control group.

5. Were the glioma tissue homogenates from the same patients as plasma samples?

6. The narrative style is visible in the rest of the text, and that aspect should be reviewed (e.g.  line 216-217: "to determine which exactly there are statistically significant differences" - that and the next sentence can be combined avoiding repetitions).

7. The visual presentation of Figure 2 could be improved, e.g. parallel graphs regarding the same protein, with the replacement of Polish words (Mediana). The same for Figure 3, 5, 6, 7, 8.

8. The use of italics in Table 3 and 5 is not understandable.

9. The Results/Discussion should be reviewed to increase precision and relation to the Figures/Tables, e.g.:

- line 382-384: I couldn't identify the data showing the described relation.

- line 399-400: If that conclusion refers to Figure 2 it is not justified.

- line 408-409: "lower [...] in grades G1-G4, while high concentrations include G4"?

10. Conclusions should be reformulated according to actual findings.

11. References should be enriched with a few up-to-date items.

Comments on the Quality of English Language

Except for the remarks above there are some grammatical/punctuation mistakes - needs one thorough reading.
